# Er^3+^/Yb^3+^ Co-Doped Fluorotellurite Glass Fiber with Broadband Luminescence

**DOI:** 10.3390/s24165259

**Published:** 2024-08-14

**Authors:** Hepan Zhu, Weisheng Xu, Zhichao Fan, Shengchuang Bai, Peiqing Zhang, Shixun Dai, Qiuhua Nie, Xiang Shen, Rongping Wang, Xunsi Wang

**Affiliations:** 1Laboratory of Infrared Materials and Devices, Advanced Technology Research Institute, Ningbo University, Ningbo 315211, China; 15679812013@163.com (H.Z.); 15375155212@163.com (W.X.); 2211100313@nbu.edu.cn (Z.F.); baishengchuang@nbu.edu.cn (S.B.); zhangpeiqing@nbu.edu.cn (P.Z.); daishixun@nbu.edu.cn (S.D.); nieqiuhua@nbu.edu.cn (Q.N.); shenxiang@nbu.edu.cn (X.S.); wangrongping@nbu.edu.cn (R.W.); 2Zhejiang Key Laboratory of Photoelectric Materials and Devices, Ningbo 315211, China; 3Ningbo Institute of Oceanography, Ningbo 315832, China

**Keywords:** Er^3+^/Yb^3+^ co-doped, fluorotellurite glass, ASE, broadband

## Abstract

In order to address the ‘capacity crisis’ caused by the narrow bandwidth of the current C band and the demand for wide-spectrum sensing sources and tunable fiber lasers, a broadband luminescence covering the C + L bands using Er^3+^/Yb^3+^ co-doped fluorotellurite glass fiber is investigated in this paper. The optimal doping concentrations in the glass host were determined based on the intensity, lifetime, and full width at half maximum (FWHM) of the fluorescence centered at 1.5 µm, which were found to be 1.5 mol% Er_2_O_3_ and 3 mol% Yb_2_O_3_. We also systematically investigated this in terms of optical absorption spectra, absorption and emission cross-sections, gain coefficients, Judd–Ofelt parameters, and up-conversion fluorescence. The energy transfer (ET) mechanism between the high concentrations of Er^3+^ and Yb^3+^ was summarized. In addition, a step-indexed fiber was prepared based on these fluorotellurite glasses, and a wide bandwidth of ~112.5 nm (covering the C + L bands from 1505.1 to 1617.6 nm) at 3 dB for the amplified spontaneous emission (ASE) spectra has been observed at a fiber length of 0.57 m, which is the widest bandwidth among all the reports based on tellurite glass. Therefore, this kind of Er^3+^/Yb^3+^ co-doped fluorotellurite glass fiber has great potential for developing broadband C + L band amplifiers, ultra-wide fiber sources for sensing, and tunable fiber lasers.

## 1. Introduction

In the optical fiber communication, the narrow bandwidth of the current C band (1530~1565 nm) is causing a ‘capacity crisis’ due to growing demand for bandwidth [1,2,3]. In 2012, Ellis stated that the communication capacity is directly proportional to the bandwidth [4]. Communication capacity can be increased by expanding the wavelength range from the C band to the L band (1565–1625 nm). Currently, erbium-doped fiber amplifiers (EDFAs) have been widely utilized as core devices in communication systems. Commercial EDFAs are made of silicate glass, and their narrow gain bandwidth, limited to the C band, has hindered the development of communication technology [2,3]. Therefore, researchers have developed various types of glass to broaden the emission bandwidth in recent years, including phosphate, oxyfluoride, tellurite, and germanate [5,6,7,8]. Tellurite glass has been studied for its excellent high rare earth doping concentration, high refractive index, lower phonon energy, excellent physical and chemical stability, and non-uniform spectral broadening due to multiple structural units (TeO_4_, TeO_3+δ_, TeO_3_) [3,9,10,11,12]. TeO_2_-BaF_2_-Y_2_O_3_ (TBY) fluorotellurite glass has a higher thermal stability and lower hydroxyl impurity than tellurite, which is advantageous for high power outputs and strong emissions at 1.5 μm [13,14,15].

It is well known that the spectral bandwidth of ASEs in glass fibers affects the gain bandwidth of fiber amplifiers [16,17]. When broadband signal light is introduced into such fibers, it can be amplified through stimulated emissions. Recent studies have highlighted the use of fibers with broadband ASE sources in fiber optic sensing systems, such as fiber Bragg grating (FBG) sensors and fiber optic gyroscopes, as well as in tunable fiber lasers [18,19,20]. Extensive research has focused on rare earth-doped fibers to enhance ASE performance [21,22]. Regarding research on the ASE broadband properties of tellurite glass fiber, Chen et al. investigated the broadband characteristics of Er^3+^-doped tellurite glass at 1.5 μm in 2006 [23]. They obtained an ASE spectrum bandwidth of 70 nm at a 3 dB intensity in a 6 m-long fiber test. Recent studies have shown that co-doping Er^3+^ with sensitizing ions such as Yb^3+^ and Ce^3+^ can further broaden the bandwidth [24,25,26,27]. In 2008, Nandi et al. tested the ASE spectrum of Er-Yb co-doped phosphor-tellurite glass fiber in the wavelength region of 1450~1650 nm, and the 3 dB bandwidth was 77 nm (1563–1640 nm) at a fiber length of 25 cm [28]. The doping concentration of Er_2_O_3_ and Yb_2_O_3_ (both 0.25 mol%) is too low to widen the spectrum. In recent years, it has been proven that increasing the concentration of Er^3+^ and Yb^3+^ can lead to a non-uniform broadening of the bandwidth [11,24,26,27]. In 2014, Desirena et al. measured an ASE spectrum width of 108 nm for Er^3+^ (Er_2_O_3_: 10,000 ppm)/Yb^3+^ (Yb_2_O_3_: 20,000 ppm) co-doped tungsten-tellurite glass fiber in an 8 cm fiber, with a high loss of approximately 11 dB/m at 1310 nm [29]. The energy from the pump and signals can be absorbed, generating significant heat, which can negatively impact the stability of the amplifier and sensor. Currently, rare earth-doped glass fibers typically experience significant loss. In 2008, Dai et al. prepared Er^3+^ (Er_2_O_3_: 0.2 mol%)/Yb^3+^ (Yb_2_O_3_: 0.4 mol%) co-doped tellurite glass with a loss of 2.1 dB/m at 1310 nm [30]. In 2016, Tang et al. prepared a Er^3+^ (Er_2_O_3_: 1 mol%)/Yb^3+^ (Yb_2_O_3_: 6 mol%) co-doped phosphate fiber with a loss of 7.5 dB/m at 1310 nm [31]. In 2021, Song et al. prepared a Er^3+^ (Er_2_O_3_: 0.5 mol%)/Yb^3+^ (Yb_2_O_3_: 1.25 mol%) co-doped bismuthate glass fiber with a loss of 2.1 dB/m at 1310 nm [32]. Regarding ASE broadband research in other glass fibers, in 2021, Zhang et al. tested a 75 cm-long antimony-silica glass fiber and obtained an ASE spectrum FWHM of approximately 72 nm [33]. In 2022, Alabbas et al. studied two different lengths of Er-doped hafnia-bismuth glass fibers [20]. By connecting them in series, they obtained an ASE source with a bandwidth of 57 nm. In 2023, Jakub et al. fabricated gallo-germanate dual-core optical fiber co-doped with Er^3+^ and Yb^3+^/Tm^3+^/Ho^3+^. In a 1 m-long fiber, they obtained an ASE spectrum with a 3 dB bandwidth of 28 nm at 1.5 μm [34]. In conclusion, the previously reported widest ASE spectrum of tellurite glass is only 108 nm, and the fiber loss needs to be further reduced.

In this work, a series of TBY glasses doped with Er^3+^ and Yb^3+^ were prepared by the optimized melt quenching method. The optimal doping concentration, determined by evaluating the fluorescence intensity, lifetime, and FWHM, is 1.5 mol% Er_2_O_3_ (15,000 ppm) and 3 mol% Yb_2_O_3_ (30,000 ppm). Core and cladding bulk glasses were prepared, the preform was extruded, and was then drawn into a fiber. The loss of the fiber at 1310 nm was 1.88 dB/m. The ASE spectrum of TBY glass fibers was studied and analyzed at different lengths under 976 nm laser diode (LD) forward pumping. A wide bandwidth ASE of approximately 112.5 nm (1505.1–1617.6 nm) at 3 dB was observed with a fiber length of 0.57 m. These innovative data will promote the development of optical fiber communication, optical sensing systems, and tunable fiber lasers.

## 2. Experimental Methods

### 2.1. Glass and Fiber Fabrication

The raw materials of TeO_2_, BaF_2_, Y_2_O_3_, Er_2_O_3_, and Yb_2_O_3_ in this paper are all of high-purity of >99.99%. Then, a series of 70TeO_2_-20BaF_2_-(10-x)Y_2_O_3_-xEr_2_O_3_ (TBYxE, x = 0, 0.3, 0.5, 0.7, 1.0, 1.5, 2.0, 2.5) and 70TeO_2_-(20-y)BaF_2_-8.5Y_2_O_3_-1.5Er_2_O_3_-yYb_2_O_3_ (TBYEyY, y = 1, 2, 3, 4, 5) glass samples were prepared by the optimized melt quenching method. During this process, the glass melt was continuously stirred, and high-purity oxygen was introduced at 950 °C for 0.5 h. The purpose was to achieve uniform mixing of the melt and to remove bubbles from within the glass melt. Subsequently, the melt was cast into a preheated brass mold and annealed at about 420 °C for 3 h before they were cooled to room temperature. Finally, the glasses were ground and polished for subsequent spectroscopic measurements. The majority of the glass preparation process followed the traditional melt quenching method [35].

For fiber preparation, the components of the cladding and core used were 70TeO_2_-17BaF_2_-8.5Y_2_O_3_-1.5Er_2_O_3_-3Yb_2_O_3_ and 65TeO_2_-25BaF_2_-10Y_2_O_3_, respectively. The core glass was ground and polished into a cylindrical shape with a height of 15 mm and a diameter of 9 mm, while the cladding glass was ground and polished into a cylindrical shape with a height of 15 mm and a diameter of 46 mm. The two glasses were stacked and extruded to make a preform which was then drawn into fibers through a drawing tower [18,36]. All the above experiments and the following measurements were taken at room temperature.

### 2.2. Measurements

The refractive indices of the glass samples were measured by infrared variable angle spectroscopic ellipsometry (IR-VASE MARK II, J.A. Woollam, Lincoln, NE, USA). The absorption spectra of the glass samples were measured with a UV–visible near-infrared (NIR) spectrophotometer (LAMBDA 950, PerkinElmer, Coventry, UK) in the range of 400–1700 nm. A UV–visible NIR steady-state transient fluorescence spectrometer (FLS980, Edinburgh Instruments, Edinburgh, UK) was used to test the up-conversion fluorescence spectra, NIR fluorescence spectra, and lifetime of the TBY glass samples. The preform was drawn into a step-indexed fiber through a fiber drawing tower (SG Controls, Cambridge, UK). Using the cutback technique, the propagation loss of the as-drawn fibers was measured at 1310 nm. The ASE spectra of the fibers were tested using an optical spectrum analyzer (OSA) (AQ6375Y, OKOGAWA, Tokyo, Japan).

## 3. Results and Discussion

### 3.1. Absorption Spectra and Judd–Ofelt (J-O) Analysis

Figure 1 shows the absorption spectra of TBY glass doped with Er^3+^ or Er^3+^/Yb^3+^ in the visible and NIR regions. In the figure, zYb (z = 1, 2, 3, 4, 5) indicates the variation in the Yb_2_O_3_ concentration, while the Er_2_O_3_ concentration is 1.5 mol%. Here, it can be seen that in the wavelength range of 350–1700 nm, there are 10 absorption peaks, which are, respectively, 380, 407, 452, 488, 522, 544, 654, 800, 980, and 1532 nm. They are responsible for the transitions from the ground state level ^4^I_15/2_ to the excited state levels, which are ^4^G_9/2_, ^2^H_9/2_, ^4^F_5/2_, ^4^F_7/2_, ^2^H_11/2_, ^4^S_3/2_, ^4^I_9/2_, ^4^I_11/2_, and ^4^I_13/2_, respectively. Er^3+^/Yb^3+^ co-doped glass exhibits a wider absorption band in the range of 870–1060 nm compared to Er^3+^ single-doped glass, as the absorption cross-section of Yb^3+^ is about ten times larger than that of Er^3+^. This indicates that co-doped Yb^3+^ can effectively absorb 980 nm of pump light [37], which can be matched with commercial high-power LD lasers. Furthermore, according to the Mc-Cumber theory [38], the maximum absorption cross-section of TBY1.5Er absorption spectra at 977 nm is calculated to be 1.978 × 10^−21^ cm^2^, whereas the maximum absorption cross-section at 800 nm is only 1.124 × 10^−21^ cm^2^.

According to the J-O theory [39,40], the measured absorption spectra, doping concentration, and refractive index of Er^3+^ need to be incorporated into the procedure given in Ref. [41]. Some spectral parameters related to the Er^3+^ single-doped TBY glass can be calculated, such as the intensity parameter Ω_t_ (t = 2, 4, 6), the electric dipole excursion probability A_ed_, the magnetic dipole excursion probability A_md_, the fluorescence branching ratio β, and the radiative lifetime τ_rad_. The calculated J-O parameters of the TBY1.5E glass samples are shown in Table 1, following the typical tellurite rule: Ω_2_ > Ω_4_ > Ω_6_. The root mean square deviation δ obtained from the fitting is low at 0.33 × 10^−6^, which indicates that the calculated J-O parameters are reliable. The parameter Ω_2_ is widely recognized as being related to the covalency between rare earth ions and ligands, as well as the symmetry of the glass host [37,42]. A larger Ω_2_ indicates a higher covalency of the Er-O bond and greater asymmetry in the surroundings of Er^3+^. As can be seen in Table 1, the covalency of the Er-O bond is higher and the symmetry of the Er^3+^ surroundings of the TBY1.5E is lower when compared to the other host glasses. The ratio of Ω_4_ to Ω_6_, known as the spectral quality factor, predicts the excited emission properties of laser glass materials [25,43]. A larger ratio indicates more intense laser transitions and greater emission cross-sections. Table 1 shows a higher ratio for TBY1.5E glass compared to other glasses, indicating its potential for achieving a strong laser output at ~1.5 μm. Based on the J-O parameters, A_ed_, A_md_, β, and τ_rad_ can be further calculated, as shown in Table 1.

### 3.2. Absorption and Emission Cross-Sections

The emission cross-section (*σ_emi_*(*λ*)) and absorption cross-section (*σ_abs_*(*λ*)) are important parameters for energy level transitions in rare earth ions. Since the transition Er^3+^:^4^I_15/2_→^4^I_13/2_ is a ground state–excited state transition, the calculation of *σ_abs_*(*λ*) can be performed using McCumber theory [38]:(1)$$\sigma_{abs}=\frac{2.303log(I_{0}/I)}{NL}$$
where *log*(*I*_0_/*I*) is the optical density determined from the absorption spectrum, *L* is the thickness of the test sample, and *N* is the Er^3+^ concentration. *σ_emi_*(*λ*) can be further calculated from the Mc-Cumber-derived *σ_abs_* [38]:(2)$$\sigma_{emi}\left(\lambda \right)=\sigma_{abs}\exp\left(\frac{\varepsilon -hc\lambda^{-1}}{kT}\right)$$
Here, *h* is Planck’s constant, *k* is Boltzmann’s constant, and *c* is the speed of light in a vacuum. The temperature-dependent excitation energy, denoted as *ε*, represents the net free energy required to excite an Er^3+^ ion from its ground state ^4^I_15/2_ to the energy level ^4^I_13/2_, while maintaining a constant temperature [50]. Figure 2a illustrates the absorption and emission cross-sections for TBYE3Y. The maximum absorption cross-section at 1531 nm is 6.48 × 10^−21^ cm^2^, and the maximum emission cross-section at 1534 nm is 7.44 × 10^−21^ cm^2^. This is larger than the 7.28 × 10^−21^ cm^2^ for phosphate glass, the 4.88 × 10^−21^ cm^2^ for fluoride oxide glass, and the 7.23 × 10^−21^ cm^2^ for bismuthate glass [51,52,53].

The gain coefficient can be determined using the absorption and emission cross-sections, as shown in the formula below [54]:(3)$$G\left(\lambda \right)=N[P\sigma_{emi}\left(\lambda \right)-\left(1-P\right)\sigma_{abs}\left(\lambda \right)]$$
Here, *N* is the Er^3+^ concentration and *P* represents the population inversion rate between two operating levels (^4^I_15/2_, ^4^I_13/2_). *P* can range from 0 to 1, with its value depending on the pump laser energy density. Figure 2b shows the gain coefficient of Er^3+^/Yb^3+^ co-doped glasses for different values of *P* (0, 0.2, 0.4, 0.6, 0.8, and 1.0). As the value of *P* increases, the wavelength of the maximum gain shifts toward shorter wavelengths, exhibiting the characteristics of a typical quasi-three-level system [26]. When *P* = 0.2, the gain coefficient of the glass is positive for wavelengths larger than 1594 nm, indicating a low pumping threshold and the easy realization of population inversion. The co-doped glass has a maximum gain coefficient of 4.08 cm^−1^ at 1534 nm, which is greater than that of the borosilicate (1.01 cm^−1^), calcium borotellurite (~0.6 cm^−1^), and phosphate glass (3.7 cm^−1^) [54,55,56]. In addition, the positive gain coefficient covers the optical communication windows in the S (1460–1530 nm), C, and L bands when *P* > 0.8, suggesting that Er^3+^/Yb^3+^ co-doped TBY glasses are potentially promising for broadband ASE sources.

### 3.3. NIR Spectra

#### 3.3.1. Single-Doped Er^3+^

Figure 3a illustrates the fluorescence intensity between 1350 nm and 1700 nm for various concentrations of Er_2_O_3_ under excitation by 980 nm LDs. It is observed that the peak fluorescence intensity around 1532 nm initially increases with the concentration of Er_2_O_3_ until it reaches 1.5 mol%. Beyond this point, the peak intensity decreases as the concentration of Er_2_O_3_ continues to rise. Figure 3b clearly shows the variation pattern of the fluorescence peak and FWHM with the increasing Er_2_O_3_ concentration. The non-uniform increase in FWHM with the increasing Er_2_O_3_ concentration can be attributed to significant variations in the environment and coordination numbers surrounding the Er^3+^ ions [57]. As seen in Figure 3c, the fluorescence lifetime of Er^3+^ at 1532 nm initially increases and then decreases with the increasing Er_2_O_3_ concentration. Specifically, it rises from 4.65 ms at 0.1 mol% to 4.91 ms at 0.7 mol% and subsequently declines to 3.28 ms at 2.5 mol%. The increase in fluorescence intensity and lifetime is primarily attributed to the enhanced absorption capacity for pumping as the concentration of Er_2_O_3_ increases. It enhances the population of Er^3+^ ions at the excited state energy level ^4^I_13/2_, thereby promoting the Er^3+^:^4^I_13/2_→^4^I_15/2_ transition, which in turn increases the fluorescence intensity and lifetime at 1532 nm. With further increases in the Er_2_O_3_ concentration, the phenomenon of Er^3+^ concentration quenching occurs, resulting in a reduction in the fluorescence intensity and lifetime at 1532 nm. Despite the wide FWHM of approximately 74 nm and 77 nm, respectively, that is attributed to the high concentrations of Er_2_O_3_ (2.0 and 2.5 mol%), there is a significant decrease in the fluorescence lifetime compared to the 1.5 mol% concentration (FWHM: ~69 nm, 1.5 mol%: 4.4 ms, 2.0 mol%: 3.7 ms, 2.5 mol%: 3.28 ms), alongside a decrease in the fluorescence intensity. All the results above may affect the gain multiplier and stability of the amplifier [58,59]. The high concentration also results in an increase in fiber loss. Therefore, by combining the above results, it can be determined that the optimal concentration of Er_2_O_3_ can be selected as 1.5 mol%.

#### 3.3.2. Er^3+^/Yb^3+^ Co-Doped

Figure 4a illustrates the fluorescence intensities for various concentrations of Yb_2_O_3_ in the 1350 nm to 1700 nm range under 980 nm LD excitation, while maintaining a constant Er_2_O_3_ concentration of 1.5 mol%. As observed from Figure 4a, the fluorescence intensity initially increases with the rise in the Yb_2_O_3_ concentration up to 3 mol%. Beyond this concentration, however, the fluorescence intensity gradually decreases. Clearly, it can be seen that there is a significant increase in the fluorescence intensity with the addition of Yb^3+^ to TBYE. When the Yb_2_O_3_ concentration increases to 3 mol%, the fluorescence intensity increases approximately 3.3-fold compared to that of TBY1.5E, indicating an effective energy transfer between Er^3+^ and Yb^3+^ ions. Figure 4b illustrates the variation in FWHM with the gradient of the Yb_2_O_3_ concentration. It is evident that the FWHM exhibits non-uniform broadening as the Yb_2_O_3_ concentration increases, ranging from 69.3 nm at 0 mol% Yb_2_O_3_ to 82.4 nm at 5 mol% Yb_2_O_3_. This is because the continuous doping of Yb^3+^ induces more changes in dopant sites and ligand fields within the TBYE glass frameworks. The distribution of Er^3+^ at various doping sites is enhanced due to the strong ionic similarity between Er^3+^ and Yb^3+^ [24]. Figure 4c shows the fluorescence lifetime for various concentrations of Yb_2_O_3_ while maintaining the Er_2_O_3_ concentration at a constant 1.5 mol%. As seen in Figure 3, the fluorescence lifetime increases from 4.4 ms to 6.57 ms as the Yb_2_O_3_ concentration increases up to 3 mol%, and then decreases with further increases in the Yb_2_O_3_ concentration. The enhancement of the fluorescence intensity and lifetime with Yb^3+^ doping primarily stems from the continuous decrease in the distance between Er^3+^ and Yb^3+^ ions, leading to an increased probability of ET (Yb^3+^:^2^F_5/2_→Er^3+^:^4^I_11/2_). This increases the population of Er^3+^ ions at the ^4^I_13/2_ energy level and enhances emissions at 1532 nm. With a further increase in the concentration of Yb_2_O_3_, the gradual decrease in the fluorescence intensity and lifetime at 1532 nm is primarily due to the ET between Yb^3+^ ions: Yb^3+^:^2^F_5/2_+ Yb^3+^:^2^F_7/2_→Yb^3+^:^2^F_7/2_+Yb^3+^:^2^F_5/2_, which results in the concentration quenching of Yb^3+^ [54]. From the above analysis, the optimum concentration of Yb_2_O_3_ is 3 mol%. By combining the optimal concentration of Er^3+^, the proposed optimal doping concentrations for TBY in this paper are Er_2_O_3_: 1.5 mol% and Yb_2_O_3_: 3 mol%.

### 3.4. Energy Transfer Mechanism

Based on the results and discussions above, and using the simplified energy level diagrams of Er^3+^ and Yb^3+^, Figure 5a illustrates the ET mechanism in Er^3+^/Yb^3+^ co-doped TBY glass. Upon excitation at 980 nm, Er^3+^ and Yb^3+^ ions are excited from their ground states to the excited states, corresponding to the transitions ^2^F_7/2_→^2^F_5/2_ for Yb^3+^ and ^4^I_15/2_→^4^I_13/2_ for Er^3+^. Since the absorption cross-section of Yb^3+^ ions at 980 nm is larger than that of Er^3+^ ions, and the two transitions—Yb^3+^:^2^F_5/2_→^2^F_7/2_ and Er^3+^:^4^I_11/2_→^4^I_15/2_—are strongly resonant, the ET1 process occurs as follows: Yb^3+^:^2^F_5/2_+Er^3+^:^4^I_15/2_→Yb^3+^:^2^F_7/2_+Er^3+^:^4^I_11/2_, resulting in more Er^3+^ ions at the ^4^I_11/2_ energy level. The ^4^I_13/2_ energy level can be populated through a non-radiative (NR) relaxation from the upper ^4^I_11/2_ energy level, followed by a radiative transition from ^4^I_13/2_→^4^I_15/2_, resulting in fluorescence at 1532 nm. Er^3+^ ions at the ^4^I_11/2_ energy level will also be excited to the ^4^F_7/2_ energy level through excited state absorption (ESA1), and ET2: Yb^3+^:^2^F_5/2_+Er^3+^:^4^I_11/2_→Yb^3+^:^2^F_7/2_+Er^3+^:^4^F_7/2_. Er^3+^ ions at the ^2^H_11/2_, ^4^S_3/2_, and ^4^F_9/2_ energy levels can be populated by the NR relaxation from upper ^4^F_7/2_, ^2^H_11/2_, and ^4^S_3/2_ energy levels, respectively, followed by radiative transitions of ^2^H_11/2_→^4^I_15/2_, ^4^S_3/2_→^4^I_15/2_, and ^4^F_9/2_→^4^I_15/2_, producing visible fluorescence at 524, 546, and 658 nm, respectively. These three types of light can be observed through up-conversion fluorescence spectroscopy, as shown in Figure 5b,c. It is noteworthy that the upward transition of Er^3+^ ions from the ^4^I_11/2_ energy level significantly attenuates the 1532 nm light, thereby generating and enhancing up-conversion fluorescence. Some Er^3+^ ions at the ^4^I_13/2_ energy level will be excited to the ^4^F_9/2_ energy level through ESA2 and ET3: Yb^3+^:^2^F_5/2_+Er^3+^:^4^I_13/2_→Yb^3+^:^2^F_7/2_+Er^3+^:^4^F_9/2_, thereby enhancing red light emission via the radiative transition ^4^F_9/2_→^4^I_15/2_. A part of Er^3+^ ions at the ^4^I_13/2_ energy level will also be excited to the ^4^I_9/2_ energy level through cross-relaxation (CR1): Er^3+^:^4^I_13/2_+Er^3+^:^4^I_13/2_→Er^3+^:^4^I_15/2_+Er^3+^:^4^I_9/2_. The ^4^I_11/2_ energy level can then be populated by NR relaxation from the upper ^4^I_9/2_ energy level, thus enhancing 1.5 μm emission. Furthermore, there is also a CR2: Er^3+^:^4^I_11/2_+Er^3+^:^4^I_11/2_→Er^3+^:^4^I_15/2_+Er^3+^:^4^F_7/2_, enhancing up-conversion emission.

Due to the high concentration of doping, Er^3+^ and Yb^3+^ ions are in close proximity, resulting in coordinated up-conversion (CUC) [60]. The process of CUC: 2Yb^3+^:^2^F_5/2_+Er^3+^: ^4^I_15/2_→2Yb^3+^:^2^F_7/2_+Er^3+^:^4^F_7/2_ enhances the up-conversion fluorescence, as clearly shown in Figure 5b,c. However, when the Yb^3+^ concentration exceeds 4 mol%, the quenching of up-conversion fluorescence occurs. This is due to the ET probability between Yb^3+^ ions being consistently higher than that between Yb^3+^ and Er^3+^ ions as the Yb^3+^ concentration increases. Due to the constant pumping of the 980 nm LD, the intensity of green fluorescence is greater than that of red fluorescence. For the Er^3+^ ions at the ^4^I_11/2_ energy level, their lifetimes are longer compared to those at the ^4^I_13/2_ energy level. This allows the ^4^I_11/2_ ions to absorb the pump light more efficiently (ESA1), resulting in the production of green light with higher intensity.

### 3.5. Fiber IR ASE

The refractive index is 1.875 at 1.55 μm for the fiber cladding glass and 1.899 for the core glass, resulting in a numerical aperture (NA) of approximately 0.3. The average loss of the fiber, determined by the cutback method at a wavelength of 1310 nm, is 1.88 dB/m. The result is shown in Figure 6a, and it is lower than that of the reported Er^3+^/Yb^3+^-doped phosphate glass fiber (7.5 dB/m) and Er^3+^/Yb^3+^ co-doped bismuthate glass (2.1 dB/m) [31,32]. Figure 6b shows that the fiber has a cladding diameter of 136.1 μm and a core diameter of 15.0 μm. The light spot diagram in Figure 6c indicates that the light was transmitted through the core during the test. For a standard step-indexed fiber, the normalized frequency V value is defined as follows [61]:(4)$$V=\frac{2\pi a}{\lambda }\sqrt{\left(n_{core}^{2}-n_{clad}^{2}\right)}$$
Here, *λ* is the wavelength, a is the radius of the core, and *n_core_*, *n_clad_* is the core, cladding refractive index. The calculated *V* value of this fiber is greater than 2.405 at around 1.55 μm, indicating that the Er^3+^/Yb^3+^ co-doped fiber prepared in this experiment transmits multimodally within the working range. 

Figure 7a shows the experimental setup for testing the ASE spectra of the TBY fiber. Furthermore, a 976 nm pump laser with 420 mW power is coupled into the TBY fiber, and the output signal of the fiber is coupled spatially with the OSA. The OSA then displays the shape of the ASE spectrum at 1.5 µm. For each length of fiber, measurements were taken at least 10 times, and the ASE spectrum with the highest spatial coupling efficiency, meaning the spectrum with the strongest ASE intensity, was selected. The ASE spectra for each fiber length are shown in Figure 7b. It is evident here that the emission peaks shift towards longer wavelengths as the fiber length increases. This is due to the ground state reabsorption of Er^3+^ in Er^3+^/Yb^3+^ co-doped TBY fibers (^4^I_15/2_→^4^I_13/2_) [28]. As the fiber length increases, this effect becomes more pronounced, leading to a gradual decrease in the energy distribution of shorter wavelengths. Figure 7c shows the change in the FWHM of the ASE spectrum with varying fiber lengths. And the maximum value of 112.5 nm (1505.1–1617.6 nm) occurs at a fiber length of 0.57 m, which is broader than the previously reported results for Er^3+^/Yb^3+^ co-doped phosphotellurite (77 nm) and Er^3+^/Yb^3+^ co-doped tellurite (~100, 108 nm) [18,28,29]. For shorter fibers, the measured ASE spectrum will closely resemble the fluorescence spectrum of glass, as reported in Ref. [31]. It can be speculated that the FWHM of the ASE spectrum for fibers shorter than 0.4 m will continue to decrease until it approaches the approximately 77.6 nm FWHM observed in Er^3+^/Yb^3+^ co-doped TBY glass. The FWHM of the ASE spectrum in the fibers, compared to that in glass, is attributed to intense radiative trapping among Er^3+^ ions in the tellurite fibers, leading to substantial spectral broadening [62]. Figure 7c also illustrates the variation in the peak intensity of the ASE spectrum for each fiber length, while maintaining a constant pump power of 420 mW. The peak intensity initially increases with the fiber length due to the corresponding increase in the number of Er^3+^ and Yb^3+^ ions, thereby enhancing pump absorption. The decrease in peak intensity with increasing fiber length is attributed to propagation losses and fluorescence trapping effects within the glass fiber [63]. It is evident that the optimal length of the optical fiber prepared in this experiment is ~0.57 m. Based on the above, the Er^3+^/Yb^3+^ doped TBY glass fiber has the potential to be used in fiber amplifiers, fiber sensing light sources, or tunable fiber lasers in the C + L bands.

## 4. Conclusions

A series of Er^3+^, Er^3+^/Yb^3+^ co-doped TBY glasses with varying concentrations were prepared. The fluorescence spectra, lifetime, and FWHM measured at ~1.55 μm showed that the optimal concentrations of Er_2_O_3_ and Yb_2_O_3_ were 1.5 mol% and 3 mol%, respectively. Among these compositions, the co-dopant of Yb^3+^ (TBYE3Y) significantly enhances the fluorescence intensity at ~1.55 μm, making it approximately 3.3 times greater than that achieved with TBY1.5E. And the FWHM is also further non-uniformly broadened, as well as the up-conversion fluorescence being significantly enhanced. Additionally, the ET mechanism of the highly doped Er^3+^/Yb^3+^ system is uncovered, involving ET not only between Er^3+^ and Yb^3+^, but also among Yb^3+^ ions themselves, resulting in the quenching of the Yb^3+^ concentration. Subsequently, the fiber preform was prepared by the extrusion method, and the fiber had an average loss of 1.88 dB/m at 1310 nm. The ASE spectrum of the fiber exhibited a typical red shift in peak wavelength as the fiber length increased. There was a similar tendency for both the intensity and FWHM, initially increasing and then subsequently decreasing. The results show that the widest FWHM of approximately 112.5 nm was found with a fiber length of 0.57 m. This broadband ASE spectrum has potential for the development of C + L wide band fiber amplifiers, sensing light sources, and tunable fiber lasers.

## Figures and Tables

**Figure 1 sensors-24-05259-f001:**
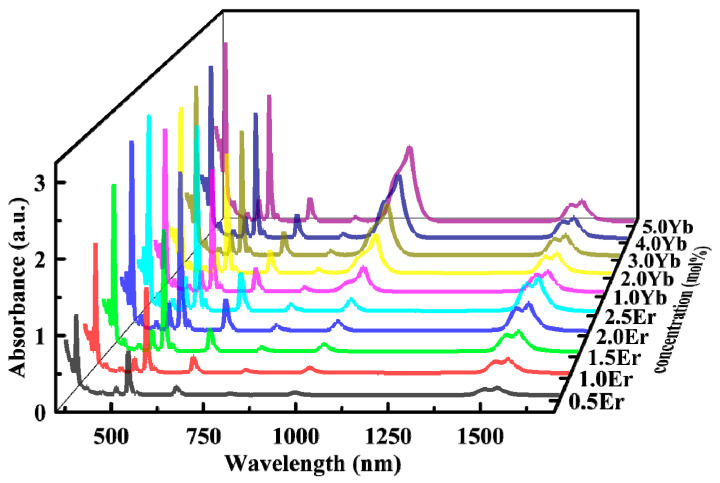
Absorption spectrum of Er^3+^ and Er^3+^/Yb^3+^ co-doped TBY glasses.

**Figure 2 sensors-24-05259-f002:**
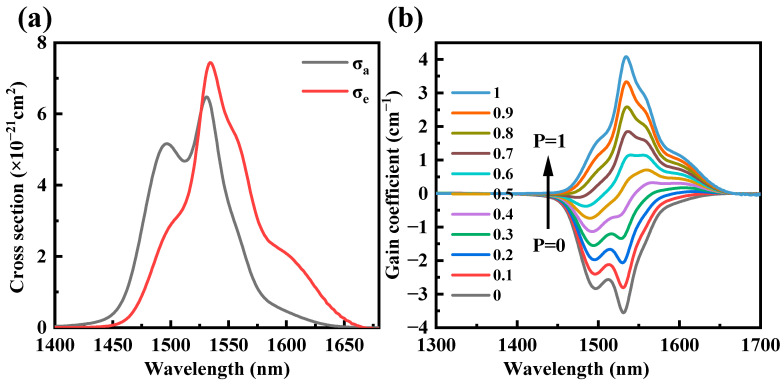
(**a**) Absorption and emission cross-sections of the Er^3+^:^4^I_13/2_←→^4^I_15/2_ transition under Er^3+^/Yb^3+^ co-doped; (**b**) Gain coefficient of the ^4^I_13/2_→^4^I_15/2_ transition for TBYE3Y.

**Figure 3 sensors-24-05259-f003:**
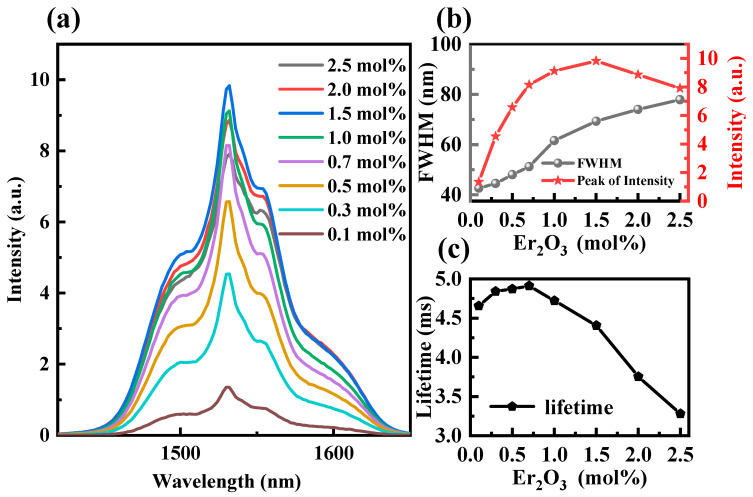
(**a**) NIR fluorescence spectra of different Er_2_O_3_ concentrations under 980 nm LD excitation; (**b**) FWHM and change in fluorescence peak under different Er_2_O_3_ concentrations; (**c**) relationship between Er_2_O_3_ concentration and lifetime.

**Figure 4 sensors-24-05259-f004:**
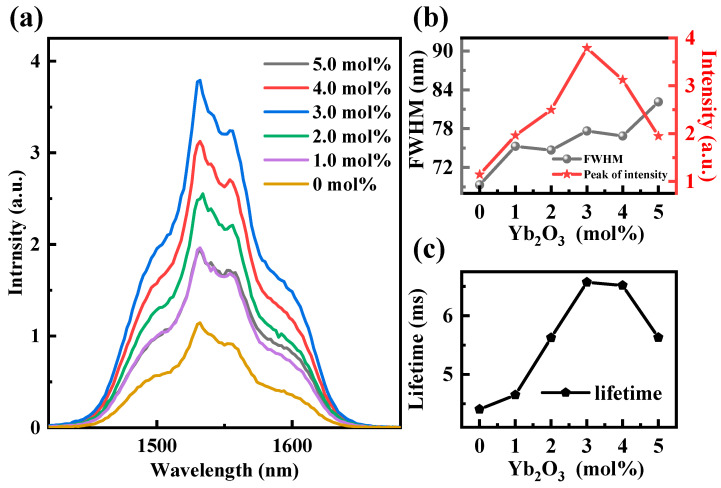
(**a**) NIR fluorescence spectra of different Yb_2_O_3_ concentrations under 980 nm LD excitation; (**b**) variation of the fluorescence peak and FWHM under different Yb_2_O_3_ concentrations; (**c**) relationship between Yb_2_O_3_ concentration and lifetime.

**Figure 5 sensors-24-05259-f005:**
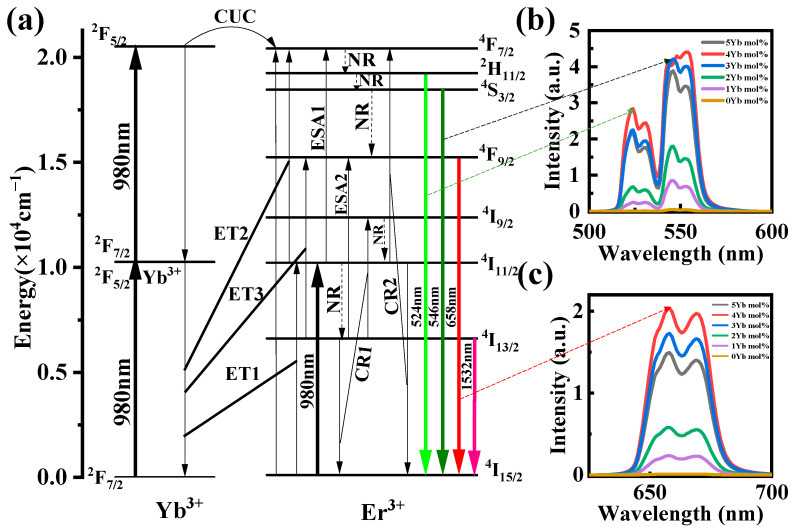
(**a**) ET diagram between Er^3+^, Yb^3+^; (**b**,**c**) Up-conversion spectra at different Yb_2_O_3_ concentrations.

**Figure 6 sensors-24-05259-f006:**
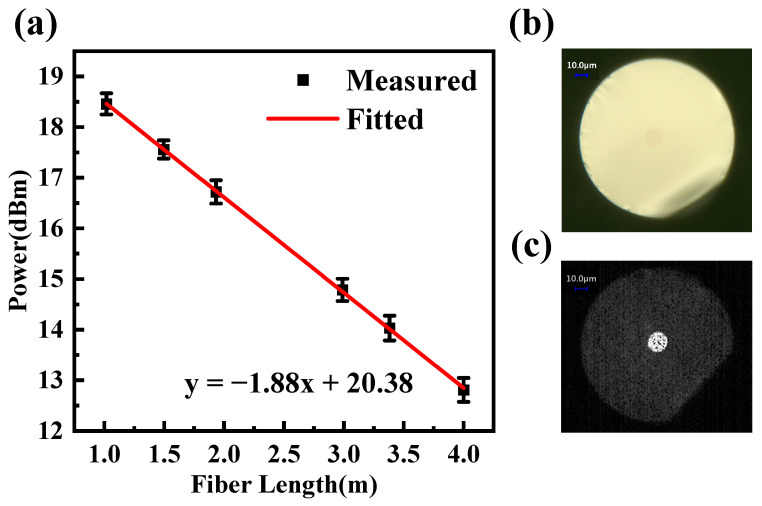
(**a**) Fiber loss test at 1310 nm; (**b**) Cross-section of fiber under optical microscope; (**c**) Spot diagram in fiber loss test.

**Figure 7 sensors-24-05259-f007:**
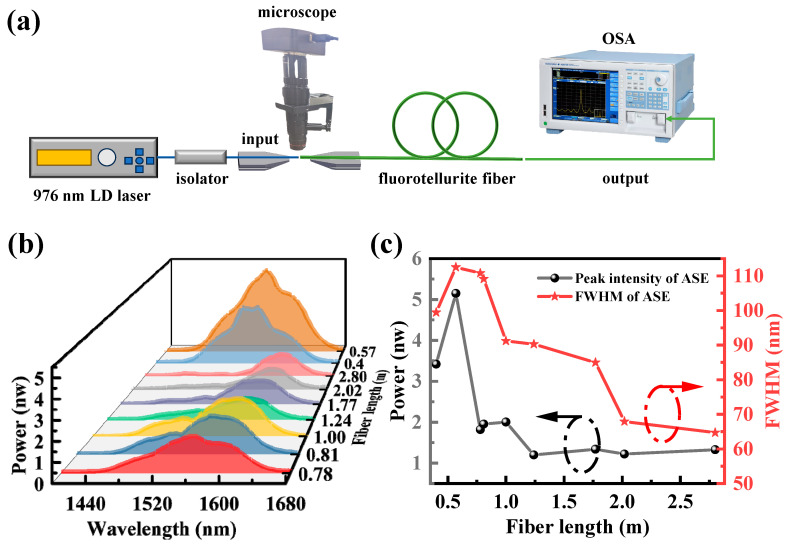
(**a**) Experimental setup for testing the ASE spectra of TBY fiber; (**b**) ASE spectrum of different fiber length; (**c**) ASE spectral FWHM and ASE spectral peak intensity of each fiber length.

**Table 1 sensors-24-05259-t001:** Electric dipole excursion probability (A_ed_), magnetic dipole excursion probability (A_md_), fluorescence branching rate (β), and radiation lifetime (τ_rad_) in Er^3+^ single-doped glasses; Judd–Ofelt intensity parameters Ω_t_ (t = 2, 4, 6) in Er^3+^ single-doped glass.

Transitions	A_ed_ (s^−1^)	A_md_ (s^−1^)	β (%)	τ_rad_ (ms)	Ω_2_ (10^−20^ cm^2^)	Ω_4_ (10^−20^ cm^2^)	Ω_6_ (10^−20^ cm^2^)	Ω_4_/Ω_6_
^4^I_13/2_→^4^I_15/2_	159.15	67.98	100	4.40	5.5	1.84	1.11	1.66 ^1^
^4^I_11/2_→^4^I_15/2_	249.96		83.54	3.34	6.09	1.74	6.34	0.27 ^2^
→^4^I_13/2_	28.19	21.05	16.46		6.17	1.5	1.10	1.36 ^3^
^4^I_9/2_→^4^I_15/2_	249.19		76.39	3.07	5.48	0.88	2.27	0.38 ^4^
→^4^I_13/2_	70.95		21.75		4.14	1.99	0.89	2.236 ^5^
→^4^I_11/2_	1.54	4.51	1.86		3.526	1.135	1.085	1.064 ^6^
^4^F_9/2_→^4^I_15/2_	2701.90		91.25	0.34	6.48	2.63	4.12	0.64 ^7^
→^4^I_13/2_	140.60		4.75					
→^4^I_11/2_	113.98		3.85					
→^4^I_9/2_	4.57		0.15					
^4^S_3/2_→^4^I_15/2_	1815.06		65.78	0.36				
→^4^I_13/2_	773.77		28.04					
→^4^I_11/2_	63.15		2.29					
→^4^I_9/2_	107.40		3.89					

^1^ This work, ^2^ oxyfluoride glass [44], ^3^ tellurite glass [45], ^4^ phosphate glass [46], ^5^ bismuth germanite glass [47], ^6^ bismuth borophosphate glass [48], ^7^ oxyhalide tellurite glass [49].

## Data Availability

The original contributions presented in the study are included in the article; further inquiries can be directed to the corresponding author.

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
