# Peer review of "Er3+/Yb3+ Co-Doped Fluorotellurite Glass Fiber with Broadband Luminescence"

_sensors, 2024, doi:10.3390/s24165259_

Round 1

Reviewer 1 Report

Comments and Suggestions for Authors

In this manuscript, a broadband luminescence of covering the C+L 15 bands using Er3+/Yb3+ co-doped fluorotellurite glass fiber is investigated. A novel doped optical fiber prerod is developed. The manuscript has the following problems that need to be corrected.

1 The references are too early, and the investigation of new materials is not enough, so the references need to be updated.

2There are no diagrams or schematics of field experiments, which makes the paper incomplete.

3Insufficient discussion on the level up degree at the peak of the spectrum.

Comments on the Quality of English Language

Moderate editing of English language required

Reviewer 2 Report

Comments and Suggestions for Authors

In this manuscript, authors prepared Er3+, Er3+/Yb3+ co-doped TBY glasses with varying concentrations, and measured their fluorescence spectra, lifetime, and FWHM at ~1.55µm to explore the optimal doping concentration. In addition, the ET mechanism of the highly doped Er3+/Yb3+ system was uncovered, and the maximum bandwidth value of the fiber and the fiber length at this value. This paper proposed a step-index fiber fluoro-tellurate glass with the widest bandwidth among all the reports based on tellurate glass, which has great potential for the development of broadband C+L band. And the innovative is clear. However, there are still some issues in this paper that need to be addressed, as follows:

1. The literature introducing the current research situation in Introduction has a long history, so it is suggested to replace or add the relevant literature in recent years to support this part.

2. It is suggested to reference relevant literature in the Glass and fiber fabrication part to increase the basis of work.

3. According to the author's description, Figure 1 shows the absorption spectrum of Er3+ and Er3+/Yb3+ co-doped TBY glasses, but the spectrum seems to give the absorption spectrum of single Er3+ and Yb3+doped TBY glasses, please explain or modify the figure.

4. It seems that the Er3+/Yb3+ co-doped TBY glasses proposed by the author is not the first idea, and it is suggested that the parameters of the relevant research should be compared in the paper to highlight the advantages and innovation of the study.

Comments on the Quality of English Language

The overall English language of the article is not much problem, but the language description can be more concise, especially the Introduction.

Reviewer 3 Report

Comments and Suggestions for Authors

The paper reports a new broadband ASE source covering the C+L bands using Er3+/Yb3+ co-doped fluorotellurite glass fiber. The broadband gain of Er3+/Yb3+ co-doped fluorotellurite glass fiber demonstrated here is impressive and the analysis is sufficient. Therefore, I recommend acceptance after a few minor revisions.

(1) The concentration values shown in Fig.1 are unclear, for the Er3+/Yb3+ co-doped TBY glasses, there is only the information of Yb concentrations, the author should also give the Er concentrations in Er3+/Yb3+ co-doped TBY glasses.

(2)  The unit mol% should be added in the legend of Fig.3-Fig.5

(3) The number of significant digits retained in the fitting result of Figure 6 should be consistent with the number of significant digits in the actual test data shown in the text.

(4)With the broadband gain, it is suggested to discuss the promising applications of the Er3+/Yb3+ co-doped fluorotellurite glass fiber beyond communications applications, such as in the high capacity sensing systems (Laser & Photonics Reviews, 2200797) and widely tunable or broadband fiber lasers (Laser & Photonics Reviews, 2400122)

Reviewer 4 Report

Comments and Suggestions for Authors

The article reported by Zhu et al. reported an ErYb co-doped fluorotellurite glass optical fiber to address bandwidth limitations. The fabricated optical fiber shows potential for broadband amplifiers and wide-spectrum sensing sources. Overall its a good piece of work and the manuscript is well written and has significant contribution in the ongoing research specifically in Amplifier. Considering the overall work the manuscript can be accepted for publication after some minor corrections listed below:

1. How many times the experiment was repeated? I must suggest to add error bars to the reported graph. What was the maximum deviation observed throughout the experiment?

2. Grammar must be checked, few typos and grammatical errors throughout the manuscript.

 3. It would be great if author can include the schematic or actual image of fabrication. How they control the uniformity of ErYb composition within the glass? Technical details must be discussed.

Comments on the Quality of English Language

Minor corrections required 
